# Natural variation in expression of a plant immune receptor mediates elicitor sensitivity

**Brian Behnken**[1], **Wesley George**[1,2], **Antonio F. Chaparro**[1,3], **Ava Kloss-Schmidt**[1,4]
**Adam D. Steinbrenner**[1]*

**1** Department of Biology, University of Washington, Seattle, Washington, United States of America,
**2** Department of Botany and Plant Sciences, University of California Riverside, Riverside, California,
United States of America, **3** Department of Plant & Microbial Biology, University of California Berkeley,
Berkeley, California, United States of America, **4** Department of Biology, New York University, New York,
New York, United States of America

☯ These authors contributed equally to this work

* astein10@uw.edu

org/10.1371/journal.pone.0343332

University, TAIWAN

**Peer Review History:** PLOS recognizes the
benefits of transparency in the peer review
process; therefore, we enable the publication
of all of the content of peer review and
author responses alongside final, published
articles. The editorial history of this article is
available here: https://doi.org/10.1371/journal.
pone.0343332

## Abstract

Plant immune systems rely on pattern recognition receptors (PRRs) to specifically
detect diverse pathogen/pest-associated molecular patterns (PAMPs). While many
distinct receptors are known to mediate PAMP recognition, the role of transcriptional
regulation of PRRs remains poorly understood. In legume plants, Inceptin Receptor
(INR) senses an 11-amino acid peptide, In11, to activate direct and indirect defenses
against caterpillar pests. Here we investigated the genetic basis of the rare In11
insensitivity phenotype found in common bean (*Phaseolus vulgaris*) landraces.
Natural variation in the rapid In11-induced ethylene response corresponded with
genetic variation at the locus encoding INR itself. Surprisingly, phenotypic variation
corresponded with expression level of *INR*, rather than coding sequence variation.
Promoter sequence variation across 21 accessions of Andean *Phaseolus vulgaris*
from northwestern Argentina, as well as near-isogenic lines (NILs) derived from
crosses between an In11-sensitive and insensitive line, corresponded with strength
of In11-induced ethylene response. Promoter alleles also corresponded with strength
of activation of a luciferase reporter in the heterologous expression model, *Nicotiana
benthamiana,* indicating that cis-element variation is sufficient to drive differences
in leaf expression levels. Surprisingly, NILs encoding either WT *INR* or the lower
expression *inr-2* allele did not show differences in resistance to herbivory by beet
armyworm (*Spodoptera exigua*), or in In11-pretreatment protection assays, sug-
gesting that even low *INR* expression can still mediate effective responses against
herbivores despite insensitivity to the In11 elicitor in laboratory assays. Our results
demonstrate that natural variation in PRR expression can contribute to differential
PAMP responses while not necessarily affecting downstream resistance phenotypes.

**Data availability statement:** Underlying data sets are available at https://github.com/BQBehnken/Argentinian_Beans_Project_Scripts We have also uploaded raw resequencing data to Zenodo at the following links: Dataset 1 (40 files) https://doi.org/10.5281/zenodo.18603657 Dataset 2 (2 files) https://doi.org/10.5281/zenodo.18636653.

**Funding:** This research was supported by NIH 5R35GM151272 and NSF 2139986 to ADS. The work was also supported by the University of Washington Royalty Research Fund, and a Washington Research Foundation Distinguished Investigator award to ADS. There was no additional external funding received for this study.

**Competing interests:** The authors have declared that no competing interests exist.

## Introduction

Plants respond to attackers through an immune system that recognizes specific pathogen and pest-associated molecular patterns (PAMPs) and activates effective defense responses. Recognition is mediated by pattern recognition receptors (PRRs) at the plasma membrane, which bind PAMPs and associate with co-receptor kinases to transduce defense signaling [1–4]. The largest group of PRRs is the set of receptor kinases (RK) and related receptor-like proteins (RLPs) encoding extracellular leucine-rich repeat (LRR) domains to bind molecular patterns [5–7]. PRRs trigger potent transcriptional reprogramming and defense responses to reduce the growth of pests and pathogens [8]; however, such responses must be regulated to avoid costs associated with growth-defense trade-offs, or otherwise coordinate optimal responses in complex environments [9–11]. Stable expression of immune receptors is therefore regulated by small RNA, epigenetic, and post-transcriptional mechanisms to maintain appropriate levels in plants [12–14].

Consistent with the need for tight regulation, different genetic models have demonstrated growth and reproductive costs of mutated or misexpressed immune receptors. In the most striking examples, immune dysregulation can occur due to specific mutations leading to autoimmunity [15]. Expression differences can also lead to subtle fitness consequences, as overexpression of individual receptors in the intracellular NOD-like receptor (NLR) family imposes reproductive costs in the absence of the pathogen, though costs depend on the specific receptor that is ectopically expressed [16,17]. Natural variation in NLR sequence or expression level can also correspond with either reduced growth or hybrid incompatibility [18,19].

In contrast to studies of natural variation in NLRs, only a few studies have explored whether PRRs that detect PAMPs mediate significant differences in resistance. Arabidopsis varieties vary in strength of response to the bacterial PAMPs flg22 (derived from bacterial flagellin) and elf18 (derived from bacterial EF-Tu) [20,21]. For elf18, two EF-Tu Receptor (EFR) haplotypes correlate with differential response to diverse elf18 ligands. However, other mapping studies in grasses studying strength of responses to flg22 and chitooligosaccharide elicitors have not implicated genetic variation in the corresponding receptors [22,23]. Natural variation of many other PRRs corresponds with strength of PAMP-triggered immunity (PTI), but it is less clear whether this variation corresponds with pathogen resistance [24–26]. Overall, how within-species natural variation at PRR loci mediates differential pest or pathogen resistance is not well understood.

To explore natural variation in a PRR-mediated immune response, we have used the Inceptin Receptor (INR) model system in legumes, which mediates recognition of herbivory. Analogous to pathogen detection, plants detect larval herbivores in the order Lepidoptera (*i.e.,* caterpillars), through the recognition of PAMPs [27]. We recently identified INR as a specific receptor for one such herbivore-derived elicitor, the inceptins, which are peptide fragments of chloroplastic ATP synthase gamma subunit that are processed and regurgitated in the foreguts of diverse caterpillar species [28–30]. The most abundant inceptin is an 11-amino acid fragment termed In11 and is present in larval oral secretions (OS) at mid-nanomolar concentrations.

Detection is mediated by the LRR-RLP Inceptin Receptor (INR) [31]. INR mediates a suite of anti-herbivore defenses, including the production of defensive phytohormones, ethylene gas, upregulation of specialized metabolites and defensive proteins, and release of herbivore-induced plant volatiles [28,29,31,32]. INR is only present in legume species in the 28-million year old lineage of Phaseoloid legumes, and thus represents a lineage-specific PRR mechanism [33]. The model system of In11 recognition by INR provides a toolkit for understanding the role of immune recognition in plant-herbivore interactions.

Among INR-encoding legume species, common bean (Phaseolus vulgaris) is especially diverse, with a history of dual domestication in both Mesoamerican and Andean regions [34,35]. Natural genetic variation within common bean therefore represents an opportunity to assess PRR evolutionary dynamics. Here, we leverage natural variation in promoter sequences across 21 varieties of common bean to explore how INR expression affects both In11 response and defense against herbivore defense.

## Results

### In11-unresponsive accessions from Argentina are genetically related

We previously identified two In11-unresponsive *Phaseolus vulgaris* accessions from USDA-GRIN germplasm by quantifying ethylene production following In11 treatment in 91 varieties [36]. Elsewhere, we described characterization of a null allele termed *inr-1*, identified in an In11-unresponsive accession from Honduras (W6 13807) [36]. A separate variety from Jujuy, Argentina (W6 17491) similarly showed a lack of In11-induced ethylene when applied to scratch-wounded leaves (w + In11) relative to wounding alone (w + $H_2O$) (Fig 1A). To further investigate this pattern, we screened 20 additional USDA-GRIN germplasm accessions from the same region of northwestern Argentina and discovered a broader subset of related landraces that were also In11-unresponsive (8 of 21 tested varieties) (Fig 1A & B, in S1 File.

To explore the potential genetic basis for In11 insensitivity among these 8 individuals, we performed low coverage genome resequencing of the 21 varieties (S2 Table in S1 File). Analysis of relatedness through single nucleotide variation indicated that unresponsive varieties were generally more related to each other than to other In-11 responsive individuals (Fig 1C & D).

### Natural variation at the *INR* promoter correlates with differential expression

To identify the potential basis for In11 insensitivity, we crossed the insensitive line PI 638858 with the genome-sequenced reference accession G19833 (S1 Fig in S1 File). $F_2$ individuals showed segregation of In11-induced ethylene response consistent with a genetic basis at a single locus, and the phenotype was fully associated with a marker single-nucleotide polymorphism (SNP) at Chr07:7,408,876, within the INR coding sequence (Fig 2A).

To analyze potential causal variation, we reanalyzed the *INR* locus (Chr07:7,407,895–7,413,204) across the 21 accessions from our WGS dataset and identified 64 SNPs at the *INR* locus. The majority of the SNPs were found in the 1.8kb promoter region (68.8%), while the rest were distributed in the coding sequence (CDS) of *INR* (S2A Fig in S1 File). We resolved six promoter haplotypes (P1-P6) spanning the −800 region upstream of *INR*. Lines responsive to In11 (13/21) predominantly possessed allele P5 with a mix of SNPs from P1, P2, and P3, while unresponsive lines (8/21) possessed mainly P2 and P4. P5 was noteworthy in that it was present in only one unresponsive individual, whereas it was widespread in the responsive individuals. Likewise, P4 was present in 5 of 8 unresponsive individuals and only one responsive individual. In summary, In11 insensitivity generally corresponded with one of two alleles (P2 and P4) based on observable SNPs in the *INR* promoter region.

In contrast, coding sequence variation did not correspond with In11 sensitivity. Three nonsynonymous CDS SNPs were observed, two isolated individuals in the leucine-rich repeat region, and one nearly fixed substitution in the C-terminus near the transmembrane domain (S2A Fig in S1 File) [38]. Despite one amino acid substitution in the island domain, a

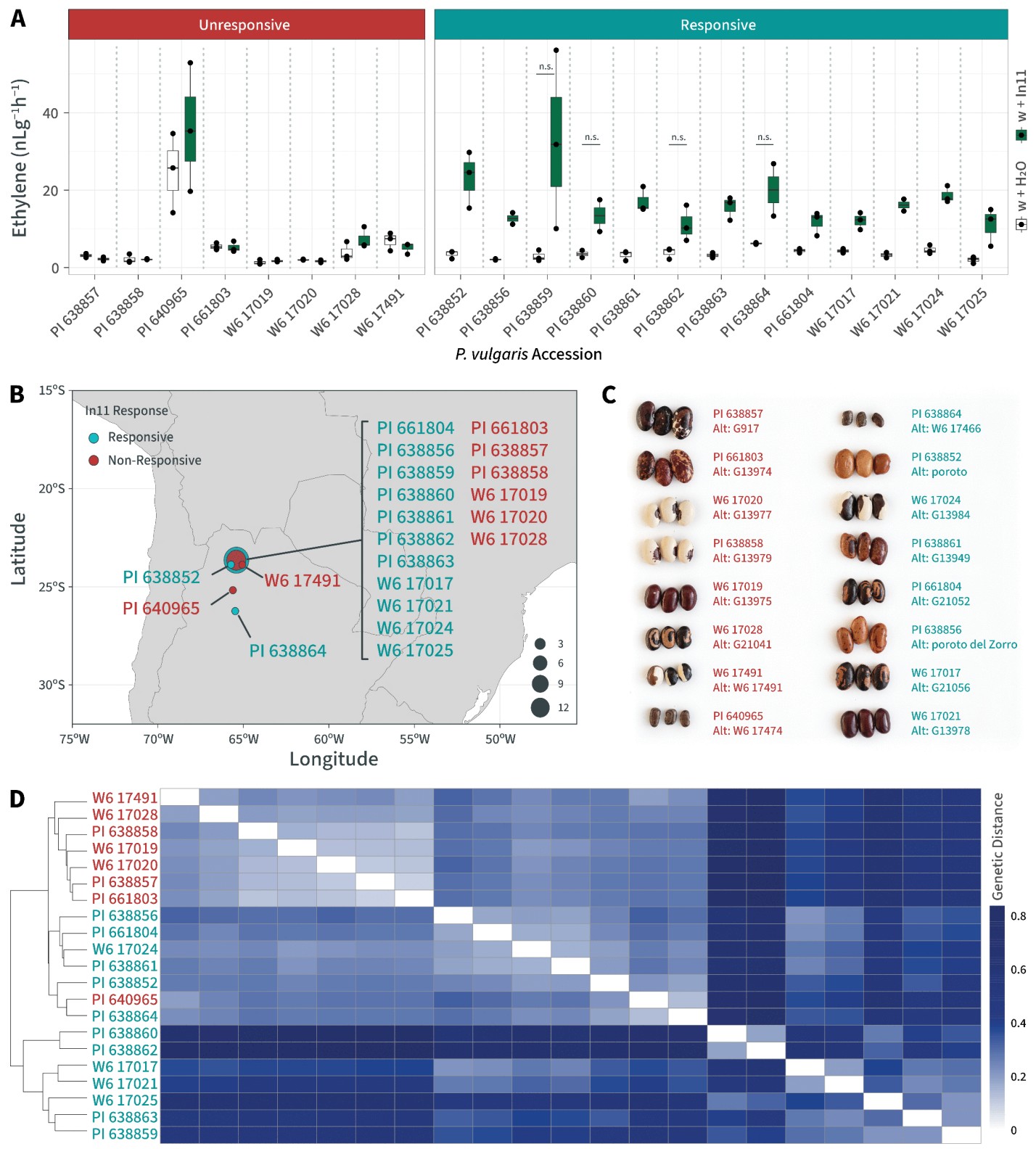

**Fig 1. In11-unresponsive accessions from Argentina are genetically related. A)** Ethylene production in Argentinian common bean accessions after scratch-wounding and treatment with $H_2O$ or 1 μM In11 peptide. Boxes show the median, 1st and 3rd quartiles and whiskers extending to the minimum and maximum values of replicate leaflets for individual In11 treatments (n = 2 or 3). Means were compared to $H_2O$ and were statistically significant for

responsive lines (Student's *t*-test, p<0.05) unless noted (n.s.), and not significant for all unresponsive lines. Responsive accessions (In11/H$_2$O ratio ≥1.9) are shown in teal, and unresponsive accessions (In11/H$_2$O ratio <1.9) are shown in red. Colors in bars represent treatment, w+H$_2$O (white) or w + In11 (green). **B)** Geographical distribution of Argentinian landraces and wild material using geotagged coordinates from USDA National Plant Germplasm System on a map of South America. Bubble size indicates number of accessions in a location. **C)** Seed coat variation is shown between select accessions. **D)** Hierarchical clustering of TASSEL (v 5.2.79) genetic distance values calculated from whole-genome single nucleotide polymorphisms (SNPs).

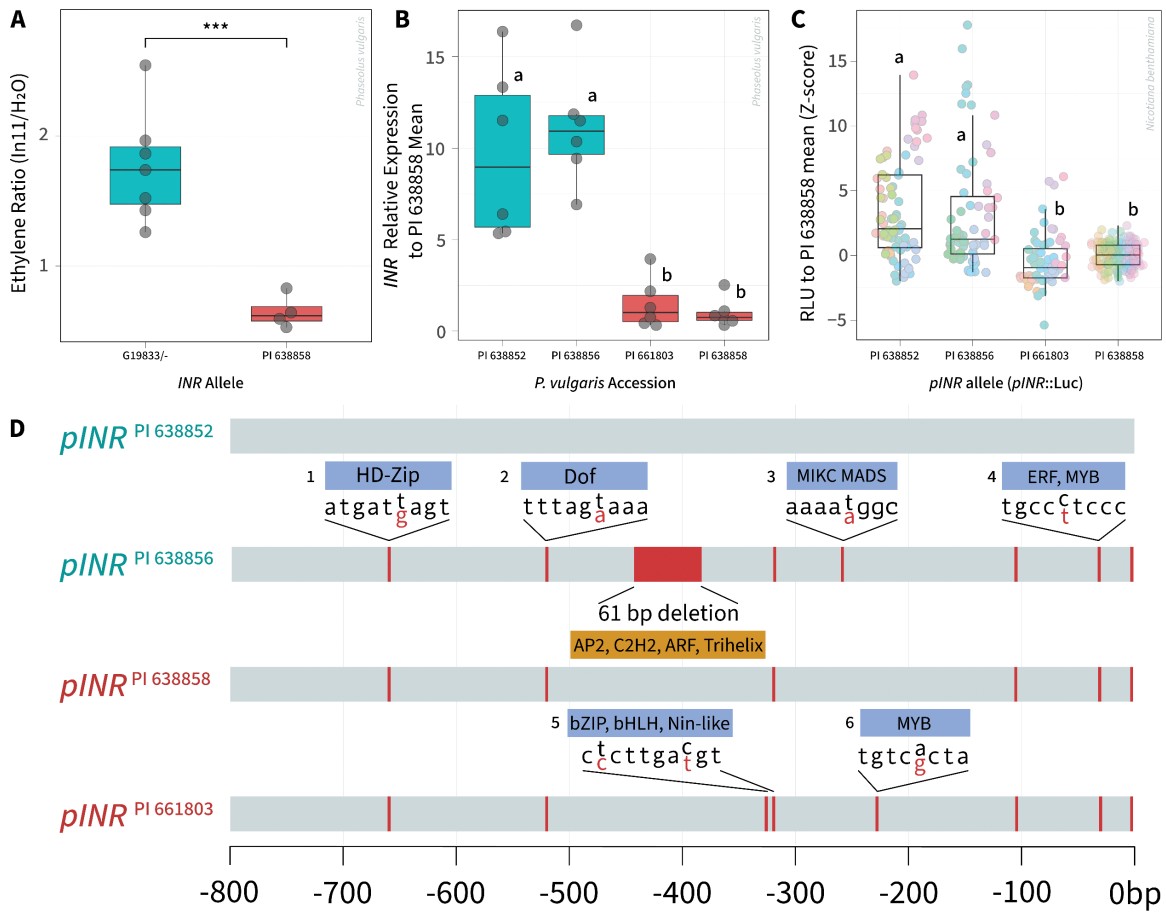

**Fig 2. Natural variation at the *INR* locus and correlation with *INR* expression and promoter strength. A)** In11 sensitivity corresponds with *INR* locus genotype in F$_2$ plants. Ethylene production after scratch-wounding and treatment with H$_2$O or 1 μM In11 peptide, with ratio greater than 1 indicating induced response. N = 11 F$_2$ plants were tested and displayed according to genotype at position 7,408,876, with plants homozygous or heterozygous for G19833 allele (G19833/-) or homozygous for PI 638858 allele (PI 638858). Boxplot shows the median, 1st and 3rd quartiles (box) and whiskers extending to the minimum and maximum values. Statistical significance between allele types was assessed by an unpaired two-tailed Student's *t*-test, with *p*-values indicated above the brackets. Significance levels: *** *p* < 0.001; ** *p* < 0.01; * *p* < 0.05; ns, not significant. **B)** Relative expression of *INR* in In11 responsive accessions from PI 638852 and PI 638856 compared to non-responsive lines PI 661803 and PI 638858. Relative expression was normalized to PI 638858 expression level. **C)** Luciferase assay data showing the Z-score of relative luminescence units (RLU) of *pINR* from common bean accessions driving luciferase relative to PI 638858 *pINR*. Colors indicate biological replicates, plotted as technical replicate leaf discs (n = 324). Different letters represent significant differences (Welch's ANOVA p < 0.05; Games-Howell post-hoc test, α = 0.05). **D)** Alignment of *pINR* sequences and Transcription Factor (TF) binding prediction of SNPs in TF binding sites upstream of the *INR* Translation Start Site (TSS) (0 bp = TSS). Only SNPs predicted to alter TF binding are shown: 1 – HD-Zip, 2 – Dof, 3 – MIKC-MADS, 4 – ERF and MYB, 5 - bZIP, bHLH, and Nin-like, 6 – MYB. TF binding was predicted with PlantRegMap parameterized with the *Arabidopsis thaliana* reference dataset [37]. Full analysis of SNPs, including those with no predicted TF binding site, is shown in S4A Fig in S1 File.

key structural component of an LRR-RLP which facilitates binding to coreceptor BAK1 [5,39], accession W6 17017 still strongly responded to In11 treatment (Fig 1A and S2A Fig in S1 File). Similarly, we detected no 3' UTR polymorphisms and therefore ruled out coding sequence and 3'-UTR variation as potential bases for In11 insensitivity.

We hypothesized that low *INR* expression from certain promoter alleles may mediate In11 insensitivity. We tested *INR* expression in six selected varieties. *INR* expression was 3.1 to 11.1-fold higher in PI 638852, W6 17024, and PI 638856 than in three unresponsive varieties PI 661803, PI 638858, and W6 17019 (Fig 2B and S3 Fig in S1 File). To test if promoter variation from these lines corresponded with In11 sensitivity, we created promoter-luciferase fusions. We selected two responsive accessions, PI 638852 and PI 638856 and two unresponsive accessions, PI 661803 and PI 638858, and tested the ability of their *pINR* alleles to drive expression in Agrobacterium-transformed *Nicotiana benthamiana*. Promoters from the two In11-responsive lines, *pINR*[PI 638852] and *pINR*[PI 638856], drove 48.0% higher luciferase activity than the *pINR*[PI 638858] from an unresponsive line, while a second promoter *pINR*[PI 661803] was similarly low in activity (Fig 2C). In summary, the strength of *INR* expression across *P. vulgaris* varieties corresponded with promoter strength in the heterologous expression system.

We compared the tested *pINR* sequences and found additional SNPs relative to those called from short-read resequencing, as well as an additional deletion from −387 to −448 bp from the TSS for *pINR*[PI 638856] (Fig 2D and S4A Fig in S1 File). Many SNPs were embedded in predicted transcription factor binding motifs. While unresponsive promoters contained several SNPs relative to the high-expression promoter *pINR*[PI 638852], none were uniquely present in unresponsive promoters. Motif enrichment analysis of these polymorphisms also did not clearly pinpoint a single transcription factor binding site (TFBS) corresponding with promoter strength (S4B-F Fig in S1 File). We conclude that a combination of promoter changes likely functions to alter expression strength in the *pINR*::*Luc* constructs.

## In11 insensitivity in low *INR* expression lines does not correspond with herbivore resistance

To test the effects of the low-expression *INR* allele, we generated recurrent backcross lines varying at the *INR* locus with a weak promoter allele as characterized above. We crossed PI 638858 with the recurrent parent G19833 for a total of four backcrosses, each time selecting for the PI 638858 allele (here termed *inr-2*), and selfed to select sibling near-isogenic lines (NILs) lines homozygous for *inr-2*/*inr-2* or the G19833 allele (*INR*/*INR*) (S1 Fig). *INR* expression level was reduced in *inr-2*/*inr-2* plants, comparable to differences between parent lines G19833 and PI 638858. (Fig 3A). In11 sensitivity as measured by In11-induced ethylene response was confirmed in *INR*/*INR* plants, while *inr-2*/*inr-2* lines remained insensitive (Fig 3B). In contrast, response to an unrelated damage-associated elicitor peptide PvPep1 [40,41] was not affected, indicating that *inr-2* specifically disrupts In11-dependent signaling. In addition, INR genotype corresponded with sensitivity of an In11-responsive marker gene encoding a MYB transcription factor previously shown to be highly induced by In11 [36], consistent with rapid In11-induced transcriptional reprogramming dependent on *INR* expression level (Fig 3C).

In order to test the effect of the *inr-2* allele on direct defenses against herbivory, we performed herbivore growth experiments using beet armyworm (*Spodoptera exigua*) on the sibling lines. No significant difference in larval weight gain between *inr-2* and *INR* plants was observed (Fig 3D). We also assessed sensitivity to In11-induced protection against herbivores, as In11 pretreatment was previously shown to reduce subsequent larval weight gain in a related legume species, cowpea (*Vigna unguiculata*) [28]. Surprisingly, pretreatment reduced caterpillar mass gain by as much as 79% for both *inr-2* and *INR* sibling lines, indicating that wounding and In11 can still elicit defense responses on *inr-2* plants in the pretreatment setup. (Fig 3D). These results indicate that In11 recognition can potentially still occur in *inr-2* lines and disrupt caterpillar physiology enough to decrease mass gain. We hypothesized that *INR* expression might increase after wounding, explaining In11 sensitivity in the pretreatment assay. However, *INR* expression was similarly low relative to PI 638856 in wounded plants at 6 h and 24 h after wounding, as well as after treatment with the damage-induced elicitor peptide PvPep1-treated plants (S5 Fig in S1 File). These results demonstrate that differences in promoter profiles differentially drive *INR* expression, potentially diminishing the immune phenotype, but unlike an *INR* null allele, the *inr-2* allele can still mediate an immune response to In11.

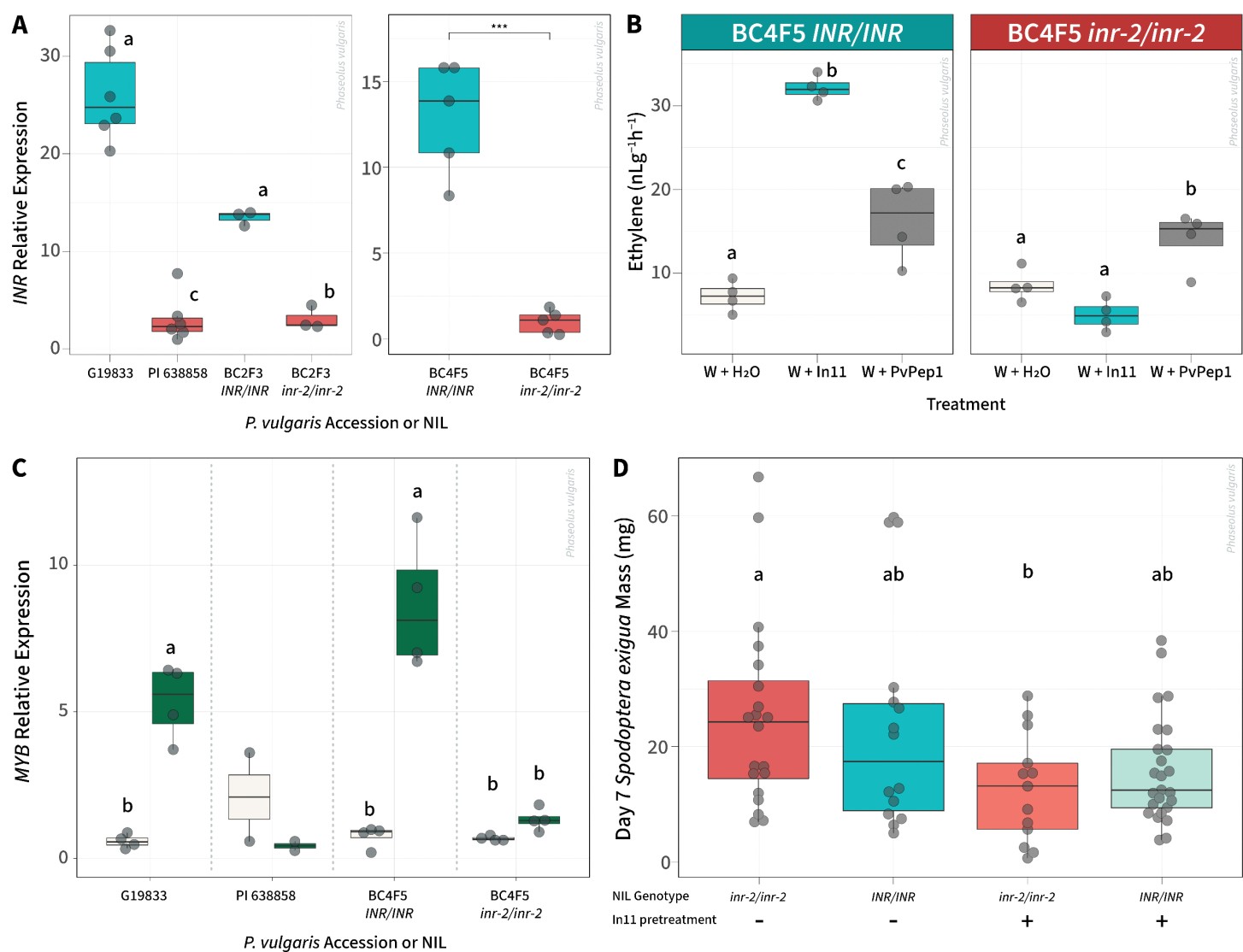

**Fig 3.** *INR* **genotype associates with lack of In11 sensitivity, but not herbivore susceptibility or In11-induced herbivore protection A) Relative expression of *INR* in the first trifoliate leaf tissue 2 weeks post germination.** Colors represent *INR* genotype in parent accessions or near isogenic lines (NILs) (G19833, *INR* vs PI 638858, *inr-2*). **B)** BC4F5 In11 response corresponds to *INR* genotype. Ethylene production in BC4F5 *INR/INR* and *inr-2/inr-2* lines after treatment with $H_2O$, 1 µM PvPep1 peptide, or 1 µm In11 peptide. Boxes show the median, 1st and 3rd quartiles and whiskers extending to the minimum and maximum values of replicate leaflets for individual water or peptide treatments (n = 6). Colors signify peptide or $H_2O$ treatment. **C)** Relative expression of *PvMYB (Phvul.001G215100)* marker gene in the primary leaf tissue. Colors in bars represent treatment, w + $H_2O$ (white), w + In11 (green). Different letters in **A)**, **B)** and **C)** represent significant differences (One-way ANOVA p < 0.05; Tukey Honestly Significant Difference (HSD), α = 0.05). Pairings that did not have letters did not reach the threshold for statistical evaluation. Statistical significance in the second facet of **A)** was assessed by an unpaired two-tailed Student's *t*-test. Significance levels: *** $p < 0.001$; ** $p < 0.01$; * $p < 0.05$; ns, not significant. **D)** Herbivore mass gain on BC4F5 plants. Final masses of 2nd instar *Spodoptera exigua* larvae feeding on common bean lines for 7 days after trifoliates were treated with inceptin-11 or left untreated. Each dot represents one larva (n = 13 - 25 per group) feeding on an individual plant. Colors represent responsive vs unresponsive lines; difference in shading represents treatment with peptide. Data is pooled from two independent experiments. Different letters indicate groups that differ significantly based on Games–Howell post-hoc comparisons (α = 0.05), following a Welch's ANOVA that indicated a significant overall effect (p < 0.05).

## Discussion

We investigated how natural variation in the promoter region of the Inceptin Receptor (INR) influences its expression and associated defense responses against herbivory in common bean (*Phaseolus vulgaris*). Using a combination of promoter characterization, advanced backcross lines, and herbivore resistance experiments across 21 varieties from Jujuy, Argentina, our findings reveal that genetic polymorphisms within the *INR* promoter correlate with receptor expression and In11 response, but that lower receptor expression does not correspond with reduced resistance to herbivores.

While we identified mutations in the *INR* promoter that might be causal variants, we were unable to identify specific cis-elements associated with In11 insensitivity. Structural differences in INR and terminator regulation hypotheses were considered and ruled out by the lack of significant polymorphisms across the 21 accessions. Instead, a TF binding site enrichment analysis of G19833 revealed a plethora of potentially herbivore-responsive sites in the −1.8kb region upstream of *INR* (S6 Fig in S1 File). Of the 6 promoter haplotypes identified in resequencing, both P2 and P4 associated with insensitivity phenotypes, and P4 drove lower expression of luciferase in the heterologous expression model. Although an initial analysis identified a specific MIKC-MADS SNP alongside another ARF/AP2 SNP that correlated with responsiveness in five responsive varieties, a mixture of cis-element variants ultimately affected expression in the heterologous model system (Fig 2D and S2 and S4 Figs in S1 File). We propose that multiple SNP combinations alter transcription factor accessibility that involve herbivore-inducible factors (bHLHs, MYBs, ERFs), as observed in other promoter studies [42,43]. Natural variation in the cis-regulatory elements of promoter regions has been previously shown to affect expression of other receptor genes [44,45] which can significantly shape adaptive responses to environmental challenges. The ability to specifically edit and study the effects of such changes in legumes will require more efficient transformation technologies.

Activation of defense pathways incurs a potential tradeoff for the plant, sacrificing growth and reproductive fecundity for short term survivability [19,46]. One hypothesis for the presence of standing promoter variation observed in In11-insensitive Jujuy accessions may reflect a cost-benefit adaptation strategy, allowing plants to conserve resources by limiting defensive responses in environments with infrequent herbivory, while still retaining the capacity to respond to sustained herbivore pressure. The Jujuy region is characterized by higher elevation, moderate temperatures (10–25ºC) and relatively low precipitation (<150mm/month) (S7 Fig in S1 File). This hypothesis potentially explains the regional proliferation of low *INR* expression genotypes.

Despite lack of In11-induced ethylene and defense gene expression in *inr-2* NILs, we showed that these lines are similarly resistant to herbivory, and that pretreatment with In11 can still protect against lepidopteran threats (Fig 3D). In contrast, a separate set of In11-insensitive NILs developed using a null receptor allele (*inr-1,* a 103-bp deletion in the INR coding sequence) increased susceptibility to herbivory [36]. Our results indicate that INR expression and In11 sensitivity alone does not predict herbivore resistance phenotypes (Figs 2B and 3D). This finding builds on recent work highlighting the complexity of elicitor-induced responses in plants, where laboratory measurements of response to a specific PAMP do not necessarily correspond with resistance [47]. We imagine several potential explanations for this incongruity. First, In11 elicitation during a natural caterpillar infestation is different from laboratory treatments; lower concentration (approximately 45 pM) In11 is present in larval OS [29], but the elicitor would be deposited over a longer period of time than in the scratch wound application protocol used here. Second, the phenotypes measured here may not reflect the full spectrum of In11-induced transcriptional and defense responses. We note that certain responses (such as induced calcium influx) are difficult to measure in common bean due to lack of genetic reporters. Third, other genetic modifiers in the recurrent parent background (variety G19833) may boost strength of In11-induced responses, overcoming the effect of *INR* expression strength. Additional NILs developed in alternate backgrounds would be needed to disentangle the effect of *inr-2* from its genetic background. Finally, defense phenotypes observed at single timepoints in growth chambers may not correspond with phenotypes in longer timecourses and more complex environmental conditions. In a recent example of this theme, natural alleles of ACD6 conferring striking differences in autoimmunity and growth phenotypes in growth chambers did not

show differences in field settings [48]. We conclude that laboratory screens based on classical, early PTI responses will likely lead to different outcomes than screens based on late-stage resistance phenotypes.

Our finding of a correlation of In11 response and *INR* expression level contrasts with mechanisms underlying other natural variation in immune responses from the literature. Examples of natural variation in pathogen resistance or PAMP response have more often been shown to correspond with presence/absence of specific receptor alleles, or allelic coding sequence variation in a receptor gene, rather than fine-tuning of expression levels [24,25,49,50]. Consistent with this, the magnitude of PAMP response does not necessarily correlate with pest and pathogen resistance in a variety of model systems [22,47,51–53]. Nevertheless, specific molecular mechanisms still govern receptor expression, suggesting that highly regulated levels of baseline receptor expression are critical for an optimized sensitivity. Given that other immune receptors vary in expression levels, and that expression level correlates with other aspects of their diversity and evolution [54,55], it would be interesting to understand if thresholds of receptor expression govern separable PTI phenotypes and pest resistance.

## Materials and methods

### Plant growth

Seeds were sown on Sunshine Mix No. 4 and germinated in reach-in PGC-FLEX growth chambers (Argus/Conviron) at 26ºC, 50% relative humidity, and 500 µMol of light with a 12/12 h light/dark photoperiod. On day twelve, plants with a fully expanded first trifoliate were transferred to a walk-in PGW40 (Conviron) at 26º, 50% relative humidity, and 500 µMol of light with a 12/12 h light/dark photoperiod prior to experiments. For *Nicotiana benthamiana*, plants were transplanted 1 week after sowing in five parts Sunshine Mix No. 4 and one part Premier Sphagnum Peat moss. Starts were fertilized with 50g Jack's Professional 20-20-20 General Purpose mixed in with 4L of water. Transplants were grown at 20°C under 12-h light and dark cycles and 160−200 µMol of light. The seedlings were grown under humidity domes for three weeks, after which the domes were removed, and the plants were grown an additional week before infiltrations. Seedlings were fertilized every other week starting at 3 weeks old with 25g Jack's Professional 20-20-20 General Purpose mixed with 4L water. Fully expanded, mature leaves of five-week-old plants were used for all transient expression experiments.

### Resequencing of *Phaseolus vulgaris* varieties

DNA was prepared from trifoliate leaflets using Macherey-Nagel Plant DNA Kit (Macherey-Nagel Inc.) Illumina libraries were synthesized and 150 bp paired-end reads were sequenced with the HiSeq X instrument by Illumina, producing ≈237.9 million paired-end read fragments in ≈68.7 Gb of data. 94.72% of reads were mapped to the *Phaseolus vulgaris* G19833 reference genome (Pvulgaris_442 v2.0) (Phvul 1.0 using BWA-MEM (v 1.2.3) [34,56] using default settings representing 100% genome coverage. The average sequence depth was 5.57-fold (2.98–9.49-fold). Genotype coverages at ≥4x, ≥ 10x and ≥20x averaged 85.89%, 62.48% and 52.55%, respectively (S2 Table in S1 File). Variants were called using Freebayes (v 0.9.14) [57]. Distance matrix was calculated using TASSEL (v 5.2.79) [58]. The TASSEL pairwise genetic distance matrix was imported into R (v 4.4.2) and clustered with average-linkage hierarchical clustering (hclust), and the resulting clustered distance heatmap and dendrogram were generated with the pheatmap package (v1.0.13) [59].

For analysis of *INR* promoter variation, a different pipeline was used. The Pvulgaris_442 v2.0 reference genome was first indexed with samtools faidx (v1.17) and the 6.8 kb *INR* interval on Chr07 (7,407,744–7,413,510) was extracted into a standalone FASTA [60]. For each accession, sorted, duplicate-marked binary alignment maps (BAMs) were then processed with bcftools (v1.17) mpileup and call (base $Q \geq 13$, $\theta = 1.1 \times 10^{-3}$) to generate VCFs, and sample-specific consensus sequences were built with bcftools consensus, masking any site with depth < 10, alternate count < 3, or QUAL < 30 for downstream alignment and SNP visualization [61]. We have marked SNP positions where there was lack of coverage in S2B Fig in S1 File. Variant and coverage data were processed in R (Version 4.4.2). Sequencing quality control statistics

were obtained using samtools and mosdepth [62] and parsed through custom R scripts using tidyverse [63]. Alignment was performed with Benchling alignment tool [64] and visualization was performed in Adobe Illustrator CC 29.7.1 and 30.0.

### Generation of *inr-2* introgression lines

PI 638858 was selected by virtue of its sexual compatibility with the *Phaseolus vulgaris* reference genome G19833. Mature stigmata from opened flowers of *Phaseolus vulgaris* PI 638858 containing the lone SNP in the promoter region of the *INR* locus (*inr-2*) were excised and used to pollinate the immature stigma from unpollinated, unopened host flowers of *Phaseolus vulgaris* G19833 that did not harbor the SNP. The resulting seeds were planted and genotyped as described. Heterozygous plants were kept and crossed with the recurrent line, G19833. This procedure was repeated until four back-cross lines were generated. For the fourth backcross line, heterozygous plants were allowed to self-pollinate and their offspring genotyped to select homozygous individuals with *INR*/*INR* or *inr-2*/*inr-2* genotypes.

### Transcription factor binding site enrichment analysis

Promoter sequences were organized into a fasta file grabbing 0 bp from the TSS to −1852 from the TSS. This fasta file was uploaded to the PlantRegMap Binding site prediction tool [37] to identify all possible TF binding sites on each sequence. TF binding sites were filtered for loci that are changed from G19833 and then annotated on an alignment chart. Alignment was done using the Benchling alignment tool [64]. Herbivore-inducible transcription factor binding site predictions were obtained using FIMO [65], part of the MEME suite. We supplied FIMO with position-specific frequency matrices for herbivore-inducible motifs and scanned the target sequence, producing a tab-delimited output file, which was imported into R (Version 4.4.2) and combined with SNP coordinates from PlantRegMap and visualized with ggplot2 [66] and Adobe Illustrator CC 29.7.1.

### Generation of Luciferase reporter constructs

*INR* promoter regions (*pINR*) of four common bean accessions (PI 638852, PI 638856, PI 661803, and PI 638858) were amplified from genomic DNA of *Phaseolus vulgaris* using gene-specific primers and using AccuPrime Taq High Fidelity MM polymerase (ThermoFisher Scientific) (S3 Table in S1 File). These primers amplified from the start codon to approximately 1.8 kb upstream. Promoter regions were then cloned into the promoter+5U acceptor backbone pICH41295 from the Golden Gate MoClo Plant Toolkit using type IIS restriction enzymes (BbsI, New England Biolabs) [67]. Constructs were selected in *E. coli* DH5α on LB agar supplemented with spectinomycin (50 µg/mL), Isopropyl β-D-thiogalactopyranoside (IPTG, 100 mM), and 5-bromo-4-chloro-3-indolyl-β-d-galactopyranoside (X-Gal, 20 mg/mL). Promoter, luciferase, and terminator L0 parts were assembled into the pGII acceptor backbone using BsaI HF®v2 restriction enzyme (New England Biolabs). Constructs were selected in *E. coli* DH5α on LB agar supplemented with kanamycin (50 µg/mL). Reporter constructs were transformed by electroporation into *Agrobacterium tumefaciens* GV3101 (pMP90, pSOUP). Constructs were selected on LB agar supplemented with kanamycin (50 µg/mL), gentamycin (50 µg/mL), rifampicin (50 µg/mL), and tetracycline (10 mg/mL). All sequences were verified by Whole Plasmid Sequencing performed by Plasmidsaurus using Oxford Nanopore Technology with custom analysis and annotation.

### Luciferase assays

*Agrobacterium tumefaciens* GV3101 harboring Luciferase reporter plasmids [67] driven by the different promoters (PI 638852, PI 638856, PI 638858, PI 661803) was cultured for 24h and then resuspended in infiltration media containing 10 mM 2-(N619 morpholino) ethanesulfonic acid (MES) (pH 5.6), 10 mM MgCl2, and 150 µM acetosyringone and diluted to a matched OD600 of 0.6 [68]. Cultures were incubated at RT for 3h in IM and then infiltrated into fully expanded leaves of

6-week-old *N. benthamiana* plants. 48h post infiltration 8 circular leaf punches (4 mm) were taken per infiltration strain per leaf and floated stomata side down on 100 μL of $H_2O$ containing 1 mM Luciferin-D in a white BioRad 96-well plate. The plate was immediately placed in a Tecan SPARK® Multimode Microplate reader following collection and relative luminescence units (RLU) were measured with 5000 ms reads per well for 90 minutes. The max RLU was taken for each well and normalized to the mean of PI 638858 bio-replicates for each test day.

### Genotyping

Samples were taken from primary leaves of common bean plants and flash frozen in liquid nitrogen. Frozen samples were then ground to a fine powder using a mixer mill (Retsch MM400, 2008, Part No. 20.745.0001) and genomic DNA extraction was performed using the NucleoSpin® Plant II kit (Machery-Nagel, 740770.520) according to the manufacturer's instructions. The concentration and quality of the extracted DNA was assessed by NanoDrop One (ThermoFisher Scientific ND-ONE-W) and gel electrophoresis. Common bean accessions were genotyped using gene-specific primers (S3 Table in S1 File) flanking a ~1000 bp region that contained a single nucleotide polymorphism (SNP) in the *PvINR* locus at position 7,408,876 in the *PvINR* locus polymorphic in the parent accessions G19833 and PI 638858. PCR reactions were performed using gene-specific primers with DreamTaq PCR MasterMix (Thermo Fisher Cat. No. K1081).

### Peptide-induced ethylene gas production and gene expression in common bean

The In11 peptide (ICDINGVCVDA) was synthesized (Genscript) based on the cATP synthase sequence from cowpea (*Vigna unguiculata*) and reconstituted in water. The PvPep1 peptide (STSLIARSGRRSTVSHGSGPQHD) was synthesized (SynPeptide) based on the *Phaseolus vulgaris* gene *Phvul.007G086700* (Phytozome).

For ethylene gas production, the left and middle leaflet of the first trifoliate or the two primary leaves of a common bean plant were lightly wounded on both sides of the main vein using a fresh razor blade to remove the cuticle, and 10 μL water, 1 μM In11, or 1 μM PvPep1 were distributed between the wounds using the tip of a pipette. After 1 h, leaflets or primary leaves were excised at the petiole using a razor blade and placed in sealed tubes for 1h before sampling 1 mL of the headspace. Ethylene gas was measured with a gas chromatograph (HP 5890 series 2, supelco #13018-U, 80/100 Hayesep Q 3FT x 1/8IN x 2.1MM nickel) with flame ionization detection and quantified using a standard curve (Scott, 99.5% ethylene, Cat. No 25881-U) [69].

For gene expression analyses, the middle leaflet of the first trifoliate or the primary leaves of a common bean plant were lightly wounded on both sides of the main vein using a fresh razor blade to remove the cuticle, and 10 μL water or 1 μM In11 were distributed between the wounds using the tip of a pipette. After 1h, leaf samples were collected by removing the treated area with a 0.6 $cm^2$ cork borer and flash frozen in liquid nitrogen. All samples were stored at −80ºC until RNA extraction.

### RNA extraction, cDNA synthesis and qRT-PCR

Total RNA was extracted using the TRIzol™ Reagent (Invitrogen, Cat. No. 15596026) method. Briefly, frozen samples were ground to a fine powder using a mixer mill (Retsch MM400, 2008), and the rest of the extraction procedure was carried out according to manufacturer's instructions. The concentration and quality of the extracted RNA was assessed by NanoDrop One (ThermoFisher Scientific ND-ONE-W) and gel electrophoresis, respectively. As DNA contamination was evident from the gel electrophoresis, the RNA was treated with TURBO DNA-free kit (Invitrogen, Cat. No AM1907) following manufacturer's instructions. cDNA was synthesized using 1 μg of RNA using the SuperScript IV Reverse Transcriptase Kit (ThermoFisher, Cat. No. 18090050). qPCR reactions were performed using gene-specific primers with PowerUp SYBR Green PCR Master Mix (Thermo Fisher, Cat. No. 4367659). Relative expression was quantified using the ΔΔCq method using ΔCq values normalized to the common bean UBQ gene and made relative to wound (S3 Table in S1 File).

## Herbivory experiments

First instar beet armyworm (*Spodoptera exigua*) larvae were obtained from Benzon Research Inc (Carlisle, PA). Upon receipt, larvae were incubated in their original packaging (artificial diet) under dark at 28ºC for 48 h to accelerate molting. Second instar larvae were weighed using a digital scale (Mettler Toledo), and ~1.5 mg caterpillars were placed in plastic cups without artificial diet. Once all masses were recorded, one single larva was placed on the first trifoliate or primary leaf of *INR/INR* and *inr-2/inr-2* plants, and leaves were enclosed in a transparent mesh bag to contain the larva. Infested plants were kept in a growth chamber at 26°C, 50% RH, and 500 µMol light, and larvae weights were recorded seven days later. For plants that were pretreated with In11, 24 hours before the experiment, the first trifoliate that the caterpillar would be placed on was scratch-wounded with a razor to remove the cuticle and 10 uL of 1 µM of In11 peptide was applied.

## Statistical analysis

All statistical analyses were conducted in R (Version 4.4.2) and manipulated using base R functions and dplyr [70]. We first tested whether residuals fulfilled the assumption of normality using a Shapiro-Wilk Test, Levene Test, and qq plots using rstatix, ggpubr, and car [71–73]. For normally distributed data, various statistical evaluations were conducted, including the Student's T-test, one-way analysis of variance (ANOVA) followed by Tukey's Honestly Significant Difference (HSD) post-hoc test (base R, multcompView [74]). In the cases where data did not conform to normal distribution and group variances, Welch's ANOVA with Games-Howell's post-hoc test (rstatix) was conducted to determine differences in relative expression between treatments or the mean mass of beet armyworm caterpillars feeding on *inr-2/inr-2* or *INR/INR* lines. Scripts used to analyze data and generate figures are available at github.com/BQBehnken/Argentinian_Beans_Project_Scripts.

## Geoanalysis

All mapping and spatial-data processing was performed in R (Version 4.4.2). The script used to generate the figures is available at github.com/BQBehnken/Argentinian_Beans_Project_Scripts. Geographic coordinates for each bean accession were retrieved from the USDA-ARS Germplasm Resources Information Network (GRIN) database and aggregated by location and response type to compute point counts. Country boundaries were obtained from rnaturalearth [75]. A bounding box was constructed around the points and passed to elevatr [76] to download ~1 km DEM data, which was converted to a terra SpatRaster [77]. Monthly mean temperature and precipitation layers (WorldClim v2.1, 10′ resolution, and 30s resolution) were read from local GeoTIFFs into two SpatRasters, cropped to the bounding box, and renamed with month abbreviations [78]. Plots were built with ggplot2 [66], exported as an encapsulated postscript (.EPS) file, and arranged in Adobe Illustrator CC 29.5.1. Visual delineation of a selection seeds from responsive and unresponsive accessions were photographed by a Nikon D750 and annotated in Adobe Photoshop CC 26.7.

## Supporting information

**S1 File. S1 Fig. Back-crossing scheme of low-expression accession into reference accession.** Back-crossing scheme of sexually compatible In11-responsive (G91833) and non-responsive (PI 638858) landraces, selecting the *INR* locus of Chr07 to generate *INR/INR* and *inr-2/inr-2* introgression lines in the G19833 background. S2 Fig. Multiple alignment of 21 accessions of Andean *Phaseolus vulgaris* at the Inceptin Receptor (*INR*) locus. A) Alignment of WGS data of *INR* (Phvul.007G077500) with vertical lines representing single nucleotide polymorphisms (SNPs) called to reference genome Pvulgaris_442 v2.1 by bcftools (v1.17) [34]. Line colors are for differentiation clarity. Each haplotype is annotated with a name (P1 - P6) and a unique color. Due to low coverage depth, haplotypes marked with an asterisk indicate a potential match with the annotated variant. Accessions annotated with asterisks indicate accessions utilized for further analysis and promoter fusion assays. B) Alignment of WGS data of *INR* promoter with SNPs that lacked sufficient

coverage marked by question marks. Vertical lines represent SNPs called by bcftools (v1.17). Low coverage is noted by dark grey bars overlaying the light grey tracks. Each haplotype is annotated with a name (P1 - P6) and a unique color matching the annotation in A). S3 Fig. *INR* expression corresponds to In11-responsive and unresponsive accessions. *INR* expression qRT-PCR data from leaf tissue in 14-day-old common beans. Expression corresponded with In11 sensitivity in responsive (PI 638852, PI 638856, W6 17024) and unresponsive accessions (PI 661803, PI 638858, W617019). Overall expression is relative to ubiquitin and normalized to undamaged, untreated tissue. S4 Fig. TF enrichment analysis did not yield single causal SNPs. A) Full 1.8 kb alignment summary and TF binding prediction from TSS (0 bp = TSS) of *INR* in common bean *INR* promoters. SNPs and deletions are indicated by red lines relative to *pINR*[PI 638858]. TF binding was predicted with the Gao labs PlantRegMap parameterized with the *Arabidopsis thaliana* reference dataset [37]. B-F) Motif enrichment analysis of 5 main SNPs that differed between accessions. Motif predictions obtained from Gao labs PlantRegMap. S5 Fig. Wounding and PvPep1 treatment do not up-regulate *INR* expression in *inr-2/inr-2* plants. qRT-PCR data from leaf tissue in 14-day-old common beans. *INR* transcriptional expression did not correspond to induction from wound or treatment with PvPep1. Overall expression is relative to ubiquitin and normalized to undamaged, untreated tissue from PI 638856 and BC4F5 *inr-2*. S6 Fig. Many potential herbivore-inducible transcription factor binding sites are predicted to be in the promoter region −1.8 kb from the TSS of *INR*. TF binding sites in *pINR* identified using FIMO, which assigns each motif occurrence a log-likelihood score (65). Higher scores indicate stronger matches to the motif model relative to background. FIMO results were merged with SNP positions obtained from PlantRegMap for visualization [37]. S7 Fig. The northwestern Andean region of Jujuy is colder, drier, and at a higher elevation A) Geographical distribution of Argentinian landraces and wild material using geotagged coordinates from USDA National Plant Germplasm System on a map of South America. Bubble size indicates the number of accessions in a location. Elevation detail of a 22.5ºS and 62ºW bounding box displayed to the right. B) Mean temperature in Cº of the same region displayed in the Elevation map in A). C) Average monthly rainfall in mm of the same region of the elevation map in A). Data for the elevation was obtained using R package elevatr, and temperature and rainfall data were obtained from WorldClim v2.1 [78]. Visualization was vector-animated using R (Version 4.4.2) and arranged in Adobe Illustrator CC 29.5.1. S1 Table. Ethylene Ratios of 21 accessions of Andean *Phaseolus vulgaris*. Ethylene response ratios of Andean *Phaseolus vulgaris* plants scratch-wounded and treated with wounding (w) + $H_2O$ or w + In11. Significance determined by Student's *t*-Test (p < 0.05). S2 Table. Summary of sequencing and mapping statistics for *Phaseolus vulgaris* accessions used in this study. Metrics include read counts, mapping rates, read quality, mismatch rates, reference genome length, mapped bases, depth of coverage, and coverage breadth at multiple thresholds. Accessions are grouped by relative *INR* expression phenotype (low vs high) according to the distance matrix from Fig 1D. S3 Table. List of primers used in this study. (ZIP)

## Acknowledgments

We thank Cora Lovell, Morgan Alonso, Jaden Benitez, and Di Wu for assistance with experiments. We thank members of the Di Stilio, Imaizumi, and Nemhauser labs for discussion.

## Author contributions

**Conceptualization:** Adam Steinbrenner.

**Data curation:** Brian Behnken, Wesley George, Antonio F. Chaparro, Ava Kloss-Schmidt, Adam Steinbrenner.

**Funding acquisition:** Adam Steinbrenner.

**Investigation:** Brian Behnken, Wesley George, Antonio F. Chaparro, Ava Kloss-Schmidt, Adam Steinbrenner.

**Writing – original draft:** Brian Behnken, Adam Steinbrenner.

**Writing – review & editing:** Brian Behnken, Wesley George, Ava Kloss-Schmidt, Adam Steinbrenner.

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
