## [Decision Letter · Decision Letter 0]

9 Oct 2025

Dear Dr. Steinbrenner,

Thank you for submitting your manuscript to PLOS ONE. After careful consideration, both reviewers gave positive feedback on this study, and we feel a minor revision and clarification would be needed before acceptance. Therefore, we invite you to submit a revised version of the manuscript that addresses the points raised during the review process.

We look forward to receiving your revised manuscript.

Kind regards,

Hao-Xun Chang, Ph.D.

Academic Editor

PLOS ONE

Journal Requirements:

https://journals.plos.org/plosone/s/file?id=ba62/PLOSOne_formatting_sample_title_authors_affiliations.pdf ..

“This research was supported by NIH 5R35GM151272 and NSF 2139986 to ADS”

“This research was supported by NIH 5R35GM151272 and NSF 2139986 to ADS”

“This research was supported by NIH 5R35GM151272 and NSF 2139986 to ADS. We thank Cora Lovell, Morgan Alonso, Jaden Benitez, and Di Wu for assistance with experiments. We thank members of the Di Stilio, Imaizumi, and Nemhauser labs for discussion.”

“This research was supported by NIH 5R35GM151272 and NSF 2139986 to ADS”

Reviewers' comments:

Reviewer's Responses to Questions

**Comments to the Author**

1. Is the manuscript technically sound, and do the data support the conclusions?

Reviewer #1: Yes

Reviewer #2: Yes

2. Has the statistical analysis been performed appropriately and rigorously?

Reviewer #1: Yes

Reviewer #2: Yes

3. Have the authors made all data underlying the findings in their manuscript fully available?

Reviewer #1: Yes

Reviewer #2: Yes

4. Is the manuscript presented in an intelligible fashion and written in standard English?

Reviewer #1: Yes

Reviewer #2: Yes

Reviewer #1: The manuscript by Behnken, George and colleagues reports an extensive genetic analysis of INR variation in common bean landraces. This work builds on the Steinbrenner group’s long-standing establishment of the ligand-receptor system as a model for herbivore-plant interactions. Focusing on an “insensitive” phenotype to the ligand (In11) treatment with the measurement of ethylene response, they identified promoter variation underlying differences in ethylene responses. They further demonstrated a significant correlation between natural variation in the promoter sequences (and thus haplotypes) and In11-sensitivity, with functional confirmation of promoter activities in the heterologous N. benthamiana system. Attempts to build a strong causative relationship have been made by generating NILs carrying the inr-2 allele. Notably, they found that the inr-2 carriers, even with low INR expression, can still confer resistance comparable to wild-type carriers. The authors provide an insightful explanation of the complex relationship between insensitivity and resistance. Their interpretation indeed deepens our understanding of how these promoter dynamics might play out in nature. Their discussion on the distinct mode evolutionary trajectories of promoter vs. genic sequences is pertinent to the new finding in this manuscript. This highlights the composite effector of various regulatory elements (e.g. TF binding sites) in finely regulating receptor function under the inherently stochastic pressures of herbivore-plant interactions.

This is an excellent and timely contribution. Rather than presenting artificially perfect datasets, it captures the complexity of natural variation and reveals the limitations of laboratory assays in fully recapitulating the forces shaping immune receptor regulation. Thus, this work provides an important gateway to improve the assay conditions to better reflect the conditions relevant for shaping the natural variation in future work.

Here, I provide minor comments that could be used for textual edits.

• L80: Is it necessary to include (LRR-RLP) as it is in the parenthesis?

• L191: The subheading can be more direct in describing the content. It is not super clear what “different” herbivore resistance means.

• Regarding the NIL-inr-2-line build-up, would the authors consider the possibility of the recurrent parent G19833 line carrying a genetic modifier (e.g. enhancing INR’s capability of conferring herbivore resistance)? It seems like the pollination/crossing of inr-2 (or other insensitive lines) seems challenging, but if there are any other NILs in different backgrounds available, the result can be reaffirmed. May the authors consider this notion as a discussion point.

• Some of the past experiments and results were condensed (e.g. L197-202). Without the knowledge of the study system, it can be a bit hard to follow. The authors can expand such parts to enhance readability.

Reviewer #2: Summary

Plant immune systems rely on pattern recognition receptors to detect diverse enemy-associated molecular patterns. The role of transcriptional regulation of these receptors is poorly understood. When functionally characterizing a genetic polymorphism for sensitivity to the herbivore-derived elicitor Inceptin in common bean, the authors serendipitously discovered that cis-regulatory variation underlies this variation. They elegantly integrated methods in plant molecular biology and classical genetics to convincingly demonstrate that cis-regulatory variation is due to promoter variation upstream of the receptor, and they found a surprising result that the organism-level immune response initiated by this receptor can be mounted to blunt herbivory even in genotypes with alleles causing the receptor to be expressed at a low level. This has fascinating implications for how plants allocate resources between growth and immunity/defense, is of broad interest, and will spur future investigations.

Overall, the manuscript is very well-written and results are presented clearly. Integration with the broader literature is deep and thoughtful: the authors put the work in context of other immune receptors to identify knowledge gaps regarding functional variation in pattern recognition receptors and the interactions between herbivores and the plant immune system.

Major Comments

None.

Minor Comments

Generally, this is written to be accessible to a broad audience. There are a couple lapses, however. For Figure 2A and related text: the experimental summaries in the results section, the figure captions, and the brief figure labels together did not clearly convey the experiment. Additionally, acronyms from molecular genetics sometimes are used without being defined (e.g., TSS). Acronyms are also used a bit heavily in general, especially in figure captions (e.g., RLU and TF in Fig. 2). Ideally, figure captions should be understandable without searching the text for definitions of acronyms. Although these acronyms may be common in certain fields, I doubt most readers will know all of them. Some extra attention to clarity of descriptions and use of acronyms would demand less effort from the reader and thus allow easier focus on the science being presented.

Fig. 1D: It might be helpful to use a distance-based clustering approach to order the accessions (keeping the same color-coding). You could either only show the tree and put the distance matrix in the supplement, or display the tree lining up with the distance matrix. At a glance, it does look like some responsive accessions could cluster in a group with the non-responsive accessions, and it would be nice to get this nuance.

Fig. 2A & L124-127: The text describes generation of F2’s from a biparental cross, which I assumed to mean standard RILs derived from that cross (i.e., as are used in QTL mapping), and references Fig. 2A. But Fig. 2A’s caption refers to something more specific: back-crossed individuals. Fig. 2A itself does not clarify this to me. I presume F2’s can be heterozygous or homozygous for either allele, so I’m not sure why the two genotypes are “G19833/-” (what does – mean?) and “PI638858” (are these only homozygous plants)? Clarification is needed.

L62: Recent work indicates that ACD6 allelic effects are highly environmentally-dependent and may be dramatically altered (to the point of not being observable) in field conditions. This environmental dependence is worth mentioning here. https://journals.plos.org/plosbiology/article?id=10.1371/journal.pbio.3003237

L102: Table S2 lists P > 0.05 for 11 species, not 8. Which of the three responsive species with P > 0.05 is not indicated in the corresponding figure? I believe the discrepancy is PI638862.

L143: I don’t understand the phrasing “the first 800 bp of 1.8 kb from TSS (0 bp = TSS)”. What is the significance of 1.8 kb? When I got to the end of the caption, I realized that perhaps 1.8 kb upstream of the TSS was analyzed for TF binding sites, but none were found more than 800 bp from the TSS? In that case, it would be clearer to simply say this rather than break it up into two non-continuous parts of the caption.

L144-145: “TF families…” is vague. Is the intended message something like, “TF families whose predicted binding sites contain SNPs and deletions are annotated”?

L298: Was fertilizer used?

L316: Given that ~15% of sites had less than 4x read depth, and low coverage sites were filtered … were there ever cases where an accession that was investigated for INR promoter SNPs did not have sufficient coverage to call the genotype at one or more SNPs? If so, it would be important to note that the genotype is uncalled rather than wildtype in the genotype plots.

L345-367: Please indicate antibiotics used for selection or growth in all cases.

L67: EFR is not defined

L80: Why is “LRR-RLP” in parentheses?

L94: For consistency, should have a dash (“In11-unresponsive”)

L95: How many In11-unresponsive accessions were identified among the 91 screened?

**Do you want your identity to be public for this peer review?** For information about this choice, including consent withdrawal, please see our For information about this choice, including consent withdrawal, please see our Privacy Policy .

Reviewer #1: **Yes:** Eunyoung ChaeEunyoung Chae

Reviewer #2: No

While revising your submission, please upload your figure files to the Preflight Analysis and Conversion Engine (PACE) digital diagnostic tool, https://pacev2.apexcovantage.com/ . PACE helps ensure that figures meet PLOS requirements. To use PACE, you must first register as a user. Registration is free. Then, login and navigate to the UPLOAD tab, where you will find detailed instructions on how to use the tool. If you encounter any issues or have any questions when using PACE, please email PLOS at . PACE helps ensure that figures meet PLOS requirements. To use PACE, you must first register as a user. Registration is free. Then, login and navigate to the UPLOAD tab, where you will find detailed instructions on how to use the tool. If you encounter any issues or have any questions when using PACE, please email PLOS at figures@plos.org . Please note that Supporting Information files do not need this step.

---

## [Author Response · Author response to Decision Letter 1]

8 Dec 2025

Please see attached Response to Reviewers word doc

---

## [Decision Letter · Decision Letter 1]

4 Feb 2026

Natural variation in expression of a plant immune receptor mediates elicitor sensitivity

PONE-D-25-47849R1

Dear Dr. Adam Steinbrenner,

We’re pleased to inform you that your manuscript has been judged scientifically suitable for publication and will be formally accepted for publication once it meets all outstanding technical requirements.

Kind regards,

Hao-Xun Chang, Ph.D.

Academic Editor

PLOS One

Additional Editor Comments (optional):

Reviewers' comments:

Reviewer's Responses to Questions

**Comments to the Author**

Reviewer #1: All comments have been addressed

Reviewer #2: All comments have been addressed

2. Is the manuscript technically sound, and do the data support the conclusions?

Reviewer #1: Yes

Reviewer #2: Yes

3. Has the statistical analysis been performed appropriately and rigorously?

Reviewer #1: Yes

Reviewer #2: Yes

4. Have the authors made all data underlying the findings in their manuscript fully available?

Reviewer #1: Yes

Reviewer #2: Yes

5. Is the manuscript presented in an intelligible fashion and written in standard English?

Reviewer #1: Yes

Reviewer #2: Yes

Reviewer #1: The revised version reads very well. I appreciate the authors' efforts to improve the manuscript according to the comments provided. Excellent work to be published soon.

Reviewer #2: All concerns have been addressed. Congratulations to the authors on a fascinating project and well-written paper!

**Do you want your identity to be public for this peer review?** For information about this choice, including consent withdrawal, please see our For information about this choice, including consent withdrawal, please see our Privacy Policy .

Reviewer #1: **Yes:** Eunyoung ChaeEunyoung Chae

Reviewer #2: No

---

## [Editor Report · Acceptance letter]

PONE-D-25-47849R1

PLOS One

Dear Dr. Steinbrenner,

I'm pleased to inform you that your manuscript has been deemed suitable for publication in PLOS One. Congratulations! Your manuscript is now being handed over to our production team.

Kind regards,

on behalf of

Dr. Hao-Xun Chang

Academic Editor

PLOS One